# comBat versus cycleGAN for multi-center MR images harmonization

**Stenzel Cackowski**[1]                    STENZEL.CACKOWSKI@UNIV-GRENOBLE-ALPES.FR

**Emmanuel L. Barbier**[1]              EMMANUEL.BARBIER@UNIV-GRENOBLE-ALPES.FR
**Michel Dojat** [1]                            MICHEL.DOJAT@UNIV-GRENOBLE-ALPES.FR
**Thomas Christen** [1]                    THOMAS.CHRISTEN@UNIV-GRENOBLE-ALPES.FR
[1] *Université Grenoble Alpes, Inserm U1216, Grenoble Institut for Neurosciences*

## Abstract

Pooling Magnetic Resonance Imaging (MRI) scans from different sites is difficult due to uncontrolled variations introduced by different acquisition protocols or scanners. Image harmonization is a way to remove site-specific bias while preserving the intrinsic image properties. While multiple harmonization techniques exist, it is yet difficult to evaluate their efficiencies in specific applications. In this paper, we propose a workflow to evaluate harmonization techniques. We carried out five experiments, performed on synthetic and real data, in order to be able to benchmark two different existing, but never compared in the literature, harmonization approaches: comBat and cycleGAN. We focus on T1-weighted MR images (one of the most widely used MR images) and propose to investigate the effects of each harmonization approach using radiomic features to extract image properties and Support Vector Machine (SVM) for classification. We show that both methods perform well for removing various types of noises while preserving manually added synthetic lesions, but also for removing site effects on data coming from 2 different sites while preserving biological information. Moreover, while each method improves autism data classification, they have different impacts on radiomic features and appear as complementary in several aspects.

**Keywords:** Brain, deep-learning, radiomic features, classification

## 1. Introduction

MR data acquired from the same patient but at different sites or scanners often lead to different MR images. This is due to the qualitative nature of the acquisitions which produces weighted images (like T1w or T2w) rather than quantitative maps. It affects the results of multi-center studies and hinders the use of publicly available online datasets such as ENIGMA, ADNI or ABIDE. It also prevents the large discrimination of machine learning tools and networks trained from a specific site may produce poor results when data are coming from different locations (Liu et al., 2020).

A possible solution to improve the results from multi-center studies is to first harmonize the data, i.e. to remove confounding site, scanner, protocol effects while retaining the biological information. Classical pre-processing steps as standardization, global scaling Fortin et al. (2017) or intensity histogram matching Shinohara et al. (2014) are not enough to counter this issue and often remove informative local variations in scans. Thus, the need for a reference harmonization technique is real. More complex techniques have been

proposed in the literature over the last decade, including Ravel (Fortin et al., 2016), comBat (Fortin et al., 2017), improved comBat versions like comBat-GAM (Pomponio et al., 2019), dictionary learning (St-Jean et al., 2020), and deep-learning methods like GANs (Dewey et al., 2019). Techniques that do not require matching subjects between sites are particularly interesting as most databases do not comprise such data. Moreover, they do not involve supplemental acquisitions and allow retrospective studies. It is however not easy to compare harmonization methods and evaluate their efficiency in the absence of ground truth.

In this study, we propose to compare 2 promising methods for the harmonization of T1-weigthed MRIs. ComBat, a statistical method already used by Fortin et al. (2017) for DTI harmonization, and cycleGAN a deep-learning model introduced by Zhu et al. (2018). In order to describe the effects of both approaches comprehensively, we ran five different experiments performed on synthetic data as well as real *in vivo* images. We first assessed the capacity of the 2 methods to remove manually added global noises in the images as well as the ability to preserve manually added lesions. We also investigated harmonization's benefits on image analysis like site classification and Autistic Syndrome Disorder (ASD) detection. Harmonization effects were evaluated using radiomic features, known to be sensitive to harmonization (Orlhac et al., 2019),(Da-ano et al., 2020)).

## 2. Materials and methods

### 2.1. Data

We used the open access ABIDE database, a multi-center project led in 2014 by Di Martino et al. (2014), focusing on autism disorders among children. It gathers more than 800 pediatric autistic patients and controls. In this study, we used healthy 3DT1-MRI scans from 2 different sites. Age range was 8-14 years old for both sites with similar sex distribution. Both acquisitions were realized on a 3T Siemens TIM trio scanner.

MR images were first co-registered to age specific 152-MNI templates publicly available (Sanchez et al., 2012). Brain was then extracted using Robex (Iglesias et al., 2011) and N4Bias (Tustison et al., 2010) was used to correct for inhomogeneities of intensity. After a manual quality check, we removed 11 scans presenting either acquisition artifacts or brain extraction issues. Finally, 51 scans were extracted for site A (56 for site B). Data was finally rescaled between [-1;1].

### 2.2. comBat

The Combined Association Test (comBat) was first introduced for reducing batch effects on genetic data (Johnson et al., 2007). It was then adapted for diffusion imaging harmonization, it exhibited a good capacity to remove unwanted site effects while preserving the desired biological (i.e. age) information (Fortin et al., 2017). (Orlhac et al., 2019) further showed comBat's efficiency for harmonizing radiomic features derived from positron emission tomography.

The comBat model can be summarized as follows. Presuming that data come from m imaging sites, with $n_i$ scans ($i = 1, 2, ..., m$). For every voxel position $v$ of scan $j$ acquired at site $i$, the intensity $y_{ijv}$ is modeled as below :

$$y_{ijv} = \alpha_v + X_{ij}\beta_v + \gamma_{iv} + \delta_{iv}\epsilon_{ijv} \tag{1}$$

Where $\alpha_v$ is the overall intensity measure for voxel $v$, $X$ is the matrix of biological covariates of interest (here age and gender) and $\beta_v$ a vector of regression coefficients corresponding to $X$ at voxel $v$. The model assumes that the error term $\epsilon_{ijv}$ follows a normal distribution $\mathcal{N}(0, \sigma^2)$. $\gamma_{iv}$ and $\delta_{iv}$ represent unwanted terms to be removed, follow normal $\mathcal{N}(\gamma_i, \tau_i^2)$ and Inverse-Gamma$(\lambda_i, \theta_i)$ distributions respectively. Model parameters are updated through empirical Bayes iterations to reduce their variance. Finally, a statistical distribution is obtained for each parameter, allowing to remove the unwanted information:

$$y_{ijv}^{comBat} = \frac{y_{ijv} - \hat{\alpha_v} - X_{ij}\hat{\beta}_v - \hat{\gamma_{iv}}}{\hat{\delta_{iv}}} + \hat{\alpha_v} + X_{ij}\hat{\beta}_v \tag{2}$$

Although it relies on a strong hypothesis for parameters priors distributions, comBat is known to be robust to small sample sizes and is considered as state of the art statistical harmonization technique for diffusion images.

### 2.3. cycleGAN

CycleGAN is a deep-learning model that can resolve Image-to-Image translation tasks (Zhu et al., 2018). The principle relies on two Generative Adversarial Models (GAN) learning how to map images translation in opposite order (GAN1 : A → B; GAN2 : B → A). When their training is successful, it is possible to recover the original input at the output of the second GAN. In the context of data harmonization, an important feature of this model is that training is unsupervised. As no ground truth is required for training, the only requirement is that the output of the second GAN matches the input of the first one. There is no need for the same subject acquisition at each site.

We used cycleGAN and developed 2D models (using axial slices) as a way to increase our sample size. Each GAN had a Pix2Pix (Isola et al., 2018) architecture with a Unet generator as presented by Ronneberger et al. (2015) and a 34*34 patch GAN as discriminator. The choice of the discriminator's field of view size was motivated by the results obtained by Modanwal et al. (2020). We used LeakyReLU activation function for the encoder part of the generators and discriminators. Classical ReLU function was used for the decoder part of the generators. Downsampling (resp. upsampling) was done through convolutional (transposed-convolutional) layers. Model loss was composed of classical Wasserstein GAN loss ($Wloss$), a l1-cycle loss consistency ($Rloss$) and a l1-loss ($Dloss$) between the input and output of each generator. This final term was found to be helpful for training and led to better convergence.

$$\mathcal{L}_{cycleGAN} = \lambda_1 Wloss + \lambda_2 Rloss + \lambda_3 Dloss$$
$$\lambda_1 = 1; \lambda_2 = 100; \lambda_3 = 3$$

Our model was trained through 1500 epochs divided in 250 steps, on batches of size 8. Learning rate was initialized to $6.10^{-4}$ and then reduced (model independently) on validation loss plateau by a factor 0.8. All training was done using Tensorflow 2.0 on a Quadro P2000 GPU / Intel Core i7-8700K CPU. A training took about 2 hours and each model was trained twice. Training procedure is detailed in A.

### 2.4. Experiments

To compare the 2 harmonization methods, we ran 3 experiments on synthetic data and 2 on real *in vivo* data (see below). Synthetic data were used to assess the ability to remove noise

or preserve known local structures. Real data were used to estimate methods efficiency to remove site effects, and their ability to improve further clinical analyses once data were harmonized.

cycleGAN was trained from scratch for each experiment. We used SVM with a radial basis function kernel to classify data before and after harmonization. To evaluate the specificity and sensitivity of our classifier, we used the Area Under Curve (AUC) of the receiver Operating Characteristic (ROC) curve. As visual inspection (C) is not enough to evaluate the effect of harmonization on the images, we extracted radiomic features known to be sensitive to site effects (Orlhac et al., 2019). We used the pyradiomics python API (van Griethuysen et al., 2017) to extract 101 features. These features aim to represent different aspects of MRIs like its shape, contrast or texture. Features are organized by families which are describe on the API's website. In all cases, we first selected the 'most correlated features' using Pearson tests (ran independently for each feature) with the characteristic of interest (site affiliation, added noise presence, etc...) using $10^{-3}$ as p-value threshold. This step was essential to focus on methods effects on characteristics of interest only. We also ran Pearson tests after harmonization on previously selected radiomic features to better understand the impact of both methods on these features. Finally, we investigated correlation between radiomic features and biological ones (sex and age). Our hypothesis was that harmonization should increase or at least preserve correlation when existing.

All results were validated by an 8 folds cross-validation. To visualize the results, we reduced the dimension with PCA and TSNE (Maaten and Hinton, 2008). PCA was first used to assure orthogonal representation of our data (8 components used, representing around 95% of total variance), and then TSNE to represent our data visually along 2 axes. Once dimensions were reduced, it was possible to visualize clusters of points corresponding to different sites or data types. For validation, we only used PCA-reduced data (8D, $> 95\%$ of variance), as there was no need for data visualization.

Finally Welch's t-tests (Welch, 1947) were run to validate if results were statistically significant or not. We ran these tests on every combination of data under the null hypotheses "method does not impact SVM accuracy" and "both methods have same performances". We then observed the p-values of these tests and rejected the $H_0$ if $p < 0.05$. Because variances of the results obtained by the two methods could not be considered as equal, we used the Welch-t test to compare whether the differences observed were statistically significant.

### 2.4.1. EXPERIMENTS 1-2: ABILITY TO REMOVE SYNTHETIC GLOBAL NOISES

We evaluated the ability of both methods to remove global noises added manually on the images. We first added Gaussian intensity shifts centered in the middle of images in order to simulate variations in MRI RF coil homogeneity. Added Gaussian noise followed a $2DNormal(0, 0.3^2)$ distribution, we then multiplied the added noise by a factor 0.6 so that the intensity in the middle of the images was increased by about 60%. Secondly, we added Gaussian noise to induce multiple artifacts and reduce contrast in the images, this was done by adding independently $\epsilon \sim N(0, 0.6^2)$ to every voxel. Removal of added noise would lead to a poor SVM accuracy for classifying the presence of noise.

### 2.4.2. Experiment 3: ability to preserve synthetic lesions

To assess that local changes in image intensities (equivalent to local lesions) were retained after harmonization, we added a localized spherical Gaussian intensity shift to some randomly taken data. Hyper-intensities preservation was estimated by SVM accuracy before and after harmonization. Moreover, we computed first order statistics (mean and variance) in the altered region to probe possible geometrical modifications due to the harmonization process. To verify that harmonization improved synthetic lesions classification, we used SVM to classify the presence of synthetic lesions using radiomic features extracted on a region of interest (where the lesion was added). CycleGAN was trained on control data only and inferences were run on all patients from site B.

### 2.4.3. Experiment 4: Site effect removal

We evaluated harmonization of the 2 selected sites through SVM accuracy as it should not be able to detect data origin. To estimate impact of harmonization on image features, we also ran Pearson tests with all radiomic features independently for site affiliation and age correlation. We looked for the number of features correlated with site affiliation after harmonization (expected to decrease), as well as features correlated with age (expected to increase).

### 2.4.4. Experiment 5: ASD patients classification

We ran a classification task (ASD patients vs healthy controls) on data from site A and B to evaluate if SVM performs better on harmonized data than on raw data. Similarly to 2.4.2 cycleGAN was trained on control data only and inferences done on all patients from site B.

## 3. Results

Table 1 (exp1-2) shows that comBat performs well on removing simple global noises, as SVM AUC metrics drop from 1 to $0.56 \pm 0.1$ while it remains close to 1 after cycleGAN. On the opposite, we show that cycleGAN performs better than comBat in the case of preserving local noises(exp3). Table 2 presents SVM AUC metrics on synthetic lesions classification. We show that, in all cases, each method does not penalize the accuracy of the SVM. For small synthetic lesions we cannot assure a benefit from both methods, whereas on larger lesions radius ($\geq 24mm$) it is clear that SVM performs better on harmonized data. We also show that in this case, cycleGAN better improves SVM performance, reaching an AUC of 1 for a radius larger than $32mm$.

For experiment 4(2.4.3) Figure 1 demonstrates the need of site harmonization as we can easily distinct two clusters representing both sites when classifying raw data (left). These clusters vanish after harmonization from both methods (all data-points are then confounded in one same cluster). Moreover, using SVM AUC metrics, Figure 2 shows that in both cases, AUC drops, attaining a minimal value (0.62 / 0.57) after cycleGAN harmonization.

Finally for experiment 5(2.4.4), Figure 3 confirms the results from previous ones. Our SVM achieves better performance on patient classification on harmonized data with the highest AUC score reached on cycleGAN harmonized data. Table 3 confirms these results

as for both experiments on site and ASD classification we have a significant improvement of SVM accuracy after both methods. Moreover, while comBat and cycleGAN show similar performances for site classification, we observe better results after cycleGAN for ASD classification.

Additionally, we show in Table Suppl.1 that each method significantly reduces the number of features correlated to site affiliation while increasing number of correlated features with age. Note that the two methods do not impact the same feature (last column), suggesting their complementarity.

| Experiment | Noise | Raw data | After comBat | After cycleGAN |
|:---:|:---|:---:|:---:|:---|
| 1 | Intensity Gaussian shift | 1 / 1 | 0.64 / **0.43** | 0.99 / 1 |
| 2 | Gaussian noise | 0.94 / 0.96 | 0.58 / **0.46** | 0.83 / 0.85 |
| 3 | Local intensity gaussian shift | 0.9 / 0.92 | 0.62 / 0.5 | 0.82 / **0.76** |

Table 1: Train / test SVM AUC for synthetic noises classification (Exp1-2-3). In bold, the best 'test' performances.

| Synthetic lesion radius | Raw data | After comBat | After cycleGAN |
|:---|:---:|:---:|:---|
| 8 | 0.61 / 0.50 | 0.68 / 0.50 | 0.77 / **0.50** |
| 16 | 0.65 / 0.50 | 0.69 / 0.50 | 0.83 / **0.50** |
| 24 | 0.76 / 0.50 | 0.79 / 0.57 | 0.91 / **0.70** |
| 32 | 0.98 / 0.83 | 0.96 / 0.75 | 0.98 / **1** |
| 40 | 1 / 1 | 1 / 1 | 1 / 1 |

Table 2: Train / test SVM AUC evolution for synthetic lesion classification (Exp3). In bold, the best 'test' performances.

| Classification | raw_data vs. comBat | raw_data vs. cycleGAN | comBat vs. cycleGAN |
|:---|:---|:---|:---|
| Site | $\mathbf{6.5 * 10^{-5}}$ | $\mathbf{2.4 * 10^{-5}}$ | 0.13 |
| ASD | $9.4 * 10^{-2}$ | $\mathbf{7 * 10^{-3}}$ | $\mathbf{4 * 10^{-2}}$ |

Table 3: P-values of Welch's t-test comparing SVM performances on different kind of data (Exp4 and Exp5). In bold significant differences of performance.

## 4. Discussion

Our results strongly support the need for harmonization and the effectiveness of comBat and cycleGAN to tackle multi-center study issues. We have shown that both methods reduced global added noises and site effects, while retaining local modifications, and improved the accuracy of the SVM for classifying synthetic lesions and ASD patients. However, it is important to point out differences between both methods performances. While comBat seems to be more adapted to remove global noises and to improve correlation between radiomic features and site affiliation or age, cycleGAN shows better results at preserving

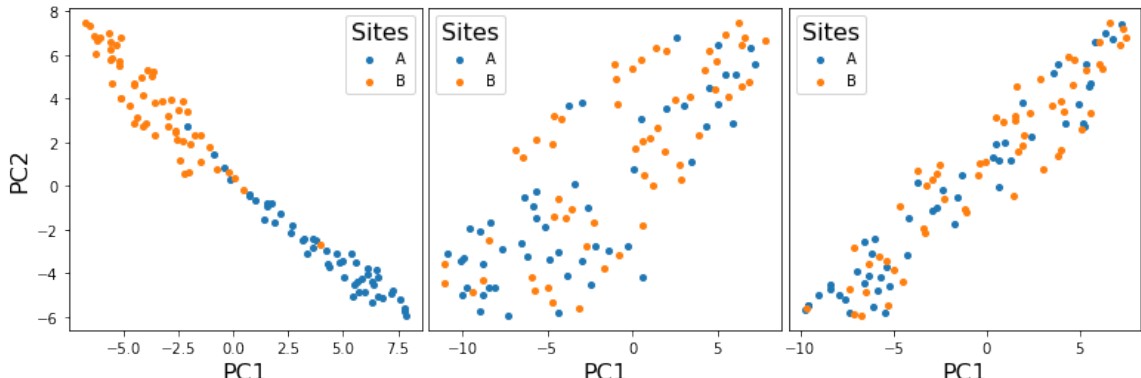

Figure 1: Sites SVM classification (Exp4), on raw data(left) and after harmonization with comBat (center) and cycleGAN (right). Corresponding classification AUC metrics are shown in Fig. 2

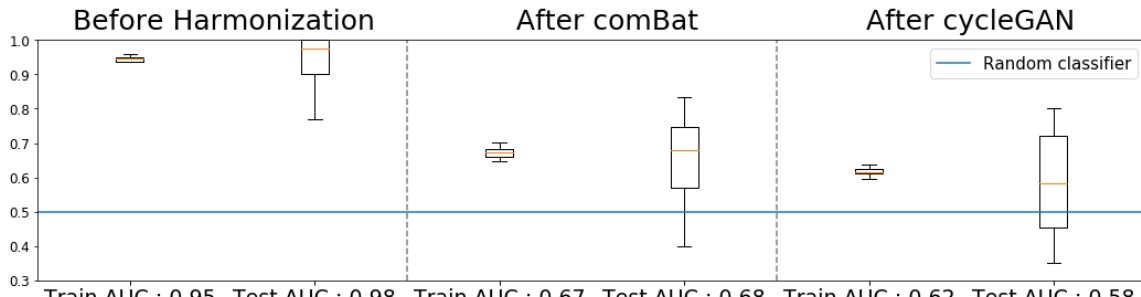

Figure 2: SVM train / test AUC metric, for site classification on control subjects (Exp4).

local modifications and at improving statistical analysis of clinical studies.

Differences between numbers of significantly correlated features in Table Suppl.1 can be explained by the fact that comBat algorithms is built to remove site affiliations effects while preserving correlations with age (as it takes age and site affiliation as inputs). On the opposite, cycleGAN only takes MRI images as input. It might be interesting to add other biological inputs like age and sex in the network to see how this affects the results of Table Suppl.1.

Giving a closer look to Pearson's tests on radiomic features, we found that both methods preserved shape-related features, as expected. The impacts of methods on other features families were found to be complementary: comBat performed well on GLRLM features while GLSZM ones benefited better from cycleGAN. Other families were similarly impacted by both algorithms. An interesting point would thus be to investigate combinations of both methods (e.g. feed cycleGAN with data harmonized by comBat, or the other way around).

Another point worth to mention is that comBat harmonizes all data when cycleGAN only modifies data from one site. This fact has an impact on our experiments. If transformations induce a noise while harmonizing, we should favor the one impacting the least images, here cycleGAN. Moreover, comBat algorithm relies on strong prior hypothesis for modeling voxels intensities, when cycleGAN tries to map the parameters directly without priors. CycleGAN also requires a much bigger sample size to be trained than comBat. (Fortin

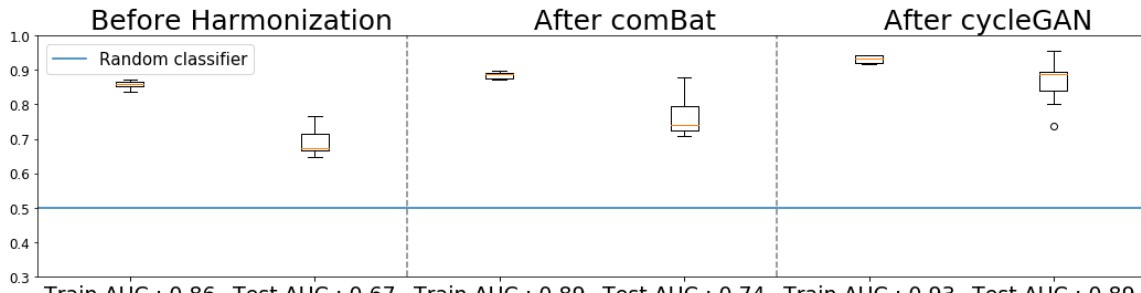

Figure 3: SVM train / test AUC metric, for ASD classification on data from sites A and B (Exp5).

et al., 2017) pointed out that comBat performs well even on small sample sizes. To illustrate this point, we also ran experiment 2.4.3 on different sites with 20 control subjects and found that cycleGAN was limited by the sample size and was not able to correct for site effects. Finally, we can point out that for each method, a new model has to be fitted for every new site encountered. This can be very time consuming and redundant, especially for cycleGAN which takes longer to be fitted than comBat. Thus, it could be very useful to investigate a way to generalize cycleGAN harmonization to every site and look for predictable features or biomarkers directly impacted by site or scanner noises.

## 5. Conclusion

In this paper we presented a workflow to evaluate harmonization techniques. We showed the importance of harmonization when dealing with data from at least 2 centers. Indeed, we were able to precisely distinguish data from two acquisition sites, even though they both used the same type of scanners. We used our workflow to investigate the performances of two harmonization algorithms for anatomical MRI multi-center studies, comBat and cycleGAN. The two aproaches can remove unwanted site effects while preserving biological information. CycleGAN results demonstrated that a deep-learning method (non linear) was well-suited for harmonization and could outperform state of the art statistical methods (linear) such as comBat in certain conditions. We could have expected that the former outperformed the latter. Surprisingly, we showed that this was not always the case. The two methods appear as complementary in several aspects and had not the same effects on radiomic features. This could determine the choice of the techniques depending on the goal to achieve. Additionally, this opens a pathway to harmonization protocols composed of several methods, such as the recently proposed Combat-GAM (Pomponio et al., 2019).

## 6. Compliance with ethical standards

This research study was conducted retrospectively using human subject data made available by the following open source: ABIDE. Ethical approval was not required as confirmed by the license attached with the data.

## 7. Acknowledgments

Stenzel Cackowski is supported by MIAI@Grenoble Alpes (ANR 19-P3IA-003). The authors have no relevant financial or non-financial interests to disclose.

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

# Appendix A. Material and Methods

---

**Protocol 1** Training-Validation-Inference protocol

---

Gather same sequence MRIs from 2 sites : OHSU & GU sites from the ABIDE database

#Prepocessing steps
Brain extraction with Robex algorithm
Co-registration to age-specific MNI templates
Intensity Biais field correction using N4 Bias algorithm
Visual quality check : brain extraction & image acquisition
Rescaling data between $[-1; 1]$
Extracting 2d axials slices while removing background slices

#Training steps $\rightarrow$ Inferring control subjects
Splitting control data in 10 folds for cross validation
**for** $i = 0; i < 10; i + +$ **do**
    Train cycleGAN using $(F_i : F_{(8+i)\%11})$ as training sets, $F_{(9+i)\%11}$ as validation set, and
    $F_{(10+i)\%11}$ as test/inference set.
**end for**

# Second training phase $\rightarrow$ Inferring ASD subjects
Gather all control subjects in one fold and ASD ones in another
Select randomly 10 control subjects for validation steps
Train cycleGAN on training control subjects
Once the model trained, run inference on ASD subjects

---

The cycleGAN architecture was preserved. The differences from the original paper come from the use of a Unet as generator (instead of a modified Resnet) and of a small DLoss term in the loss function. The rationale was that Unet tends to preserve object's shape (which is an important aspect in MRI harmonization) thanks to its skip connections. Note that this type of implementation is commonly used: for instance with TensorFlow ( https://www.tensorflow.org/tutorials/generative/cyclegan) or for dMRI harmonization using dualGAN Yi et al. (2018).
The DLoss term was added to prevent the model from well-known instabilities leading to drastic changes to the image (we expected smooth shape). This term was also helpful for training (as this kind of models are known to be hard to train). The order of magnitude of the DLoss was however small (10E-4) compared to our training loss (order of magnitude was 10E0). We thus do not expect major impact on training results.

## Appendix B. Results: Pearson's tests results

|      | Raw data | After comBat | After cycleGAN | common features |
|------|----------|--------------|----------------|-----------------|
| Site | 71       | **6**        | 19             | 3               |
| Age  | 27       | **49**       | 34             | 22              |

Table Suppl.1: Number of radiomic features significantly correlated with site affiliation and age, among the 101 extracted. Obtained with a Pearson's test. The last column corresponds to the number of significantly correlated features common to both methods.

## Appendix C. Results: Visualisation

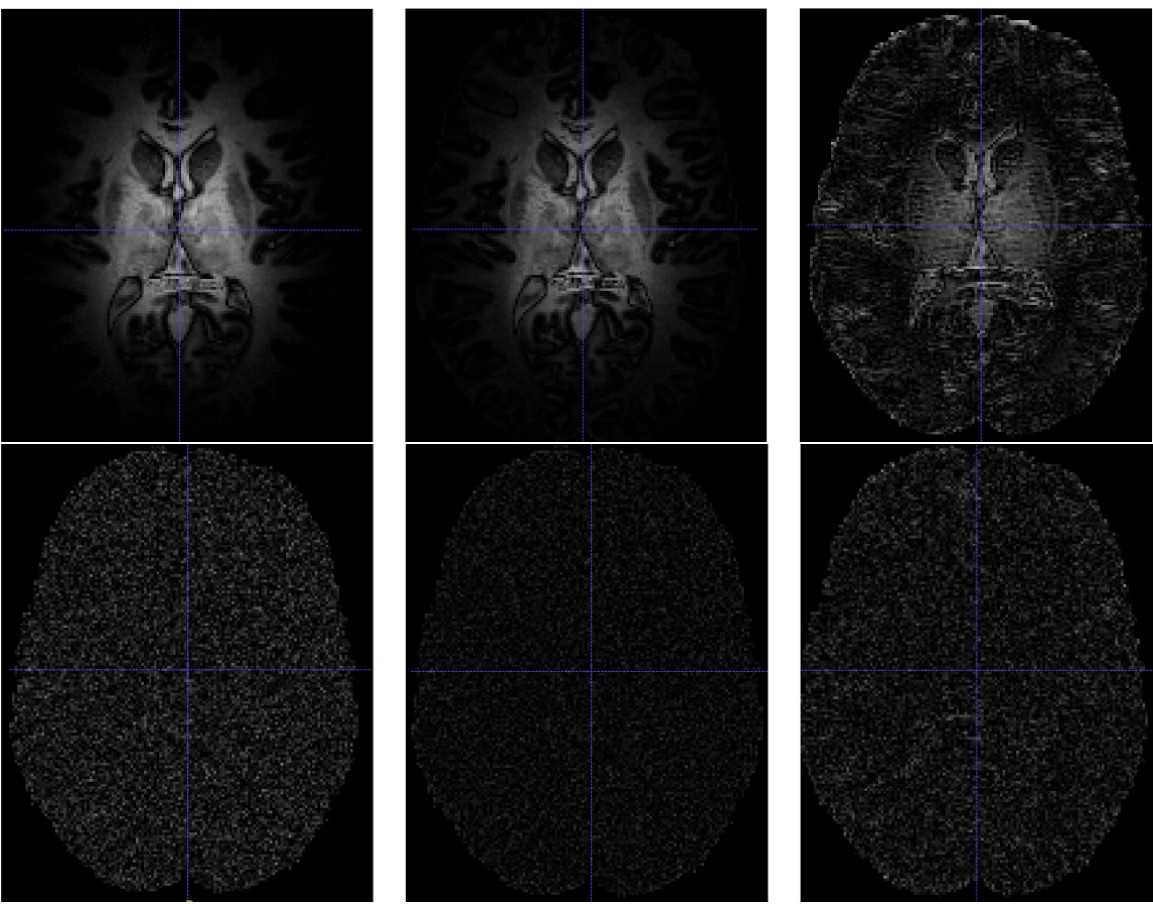

Figure Suppl.1: Method's effects on global noises 2.4.1 : Gaussian intensity shift (exp. 1)in top row, and global Gaussian noise (exp. 2) in bottom row. From left to right : Noise added to the image; absolute difference between original scan and comBat-denoised image; absolute difference between original scan and cycleGAN-denoised image

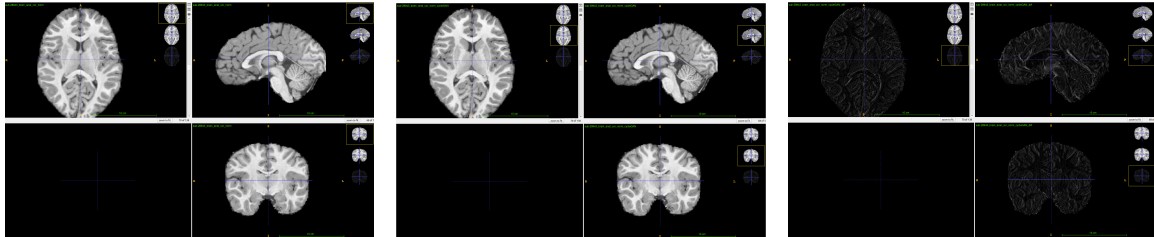

Figure Suppl.2: Impact of cycleGAN method. From left to right : Ground Truth data before harmonization; cycleGAN harmonized result; differential image showing modifications by cycleGAN for exp.4-5

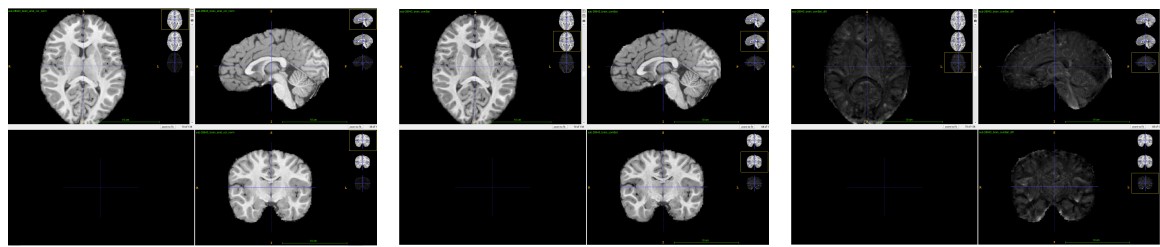

Figure Suppl.3: Impact of comBat method. From left to right : Ground Truth data before harmonization; comBat harmonized result; differential image showing modifications by comBat for exp.4-5

## Appendix D. Results: Intensity distributions

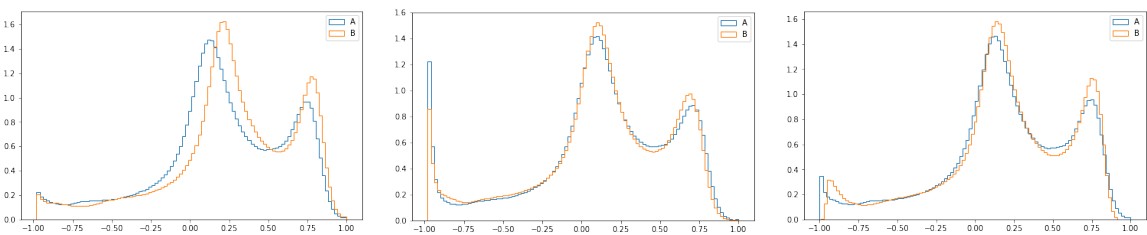

Figure Suppl.4: Intensity distribution across sites before and after both methods. From left to right : Ground Truth data; after comBat; after cycleGAN

