# OpenReview forum: "comBat versus cycleGAN for multi-center MR images harmonization"
_MIDL.io/2021/Conference — Submitted to MIDL 2021_

### Official Review · AnonReviewer1 · 2021-03-07

**Confidence:** 3
**Preliminary Rating:** 3
**Recommendation:** Poster
**Final Rating:** 3

**Summary:**

Authors present a comparison between two MR image harmonisation techniques: comBat and a CycleGAN. This is useful for larger studies including data from multiple sites, for which there is a need for harmonisation of the acquired images. Authors use five different experiments, three on synthetic data and two on real images.

**Strengths:**

The comparison of the two techniques comBat and a CycleGAN is valuable to know and of interest to the research community. Harmonising data is of increasing importance, since data set sizes keep increasing and pooling of multiple sources is becoming more used to train (deep) machine learning techniques. The five different experiments give a good overview of the techniques.

**Weaknesses:**

Authors mention a number of harmonisation techniques, but only compare two of them in their manuscript. The work would be more valuable if more methods would have been compared.

Authors use only real data originating from two sites. While this illustrates the concept, it would have been more valuable to have a larger data set with data from many more different sites.

Especially since the data originates from the same ABIDE study, it could very well be that there has already been considerable effort to harmonise the MR imaging protocols. If that has been the case (which is e.g. the case for ADNI), image harmonisation after acquisition becomes easier. Authors do not address this, but it could very well be that their good results are partially due to acquisition harmonisation efforts.

Authors only demonstrate one true clinical problem: ASD classification. Including more clinical problems into the comparison would increase the value of the work. E.g. including brain segmentation performance, lesion quantification, etc.

**Deanonymize Review:**

no

**Detailed Comments:**

The source of the synthetic data is not clear to me, please add this to the manuscript.
For the ABIDE data set, the number of controls and the number of patients is not clear to me, please add this.
Please also explain why only 51+56 subjects were used out of the 800+ available subjects.
There are quite some spelling and grammar mistakes in the work, for example:
- N4Biais
- "experiences" instead of "experiments"
- All Table / Figure words are double in the Results
- ... there are many more, I did not write them down; please go through the manuscript carefully and edit for language

2.4.1 please split into two sections, so that Experiment 1 and Experiment 2 each have their own section.
Please do not abbreviate Experiment into Exp, there is no need for that.
It would be nice if authors could include example images of the RAW data, data with noise (experiments 1 and 2), data with fake lesions (experiment 3), etc...

In the results, please repeat the Experiment numbers to make it more clear which results are presented where.

Table 1: please include a column specifying which experiment is which row.
Table 2: please make the number of significant digits the same everywhere (e.g. always 1 decimal digit)

**Final Rating Justification:**

I'm satisfied with the authors comments and I keep my original rating.

**Justification Of The Preliminary Rating:**

The work is of interest to the MIDL community, where increasing data set size for machine learning by pooling from multiple sources is a valuable strategy. This work demonstrates which techniques could be used / considered.

**Paper Type:**

validation/application paper

**Special Issue:**

no

---

### Official Review · AnonReviewer4 · 2021-03-08

**Confidence:** 5
**Preliminary Rating:** 2

**Summary:**

Paper #182
ComBAT versus cycleGAN for multi-center MR images harmonization

Summary
This paper proposes a comparison of two harmonization approaches when classifying Autism Spectrum Diseases (ASD) in ABIDE. The first, ComBAT (Jonhson et al, 2007), is based on a linear model optimized with a Bayesian approach. The second, a CycleGAN (Zhu et al, 2018), allows for non-linear modeling in the data harmonization. The evaluation consists of observing reduction of site-effects in a classification task in the presence of added noises in synthetic data, and contributions on radiomic features to a ASD classification in ABIDE.

**Strengths:**

Strengths
The paper provides a discussion on wether using comBAT or a cycleGAN is better in data harmonization. This may be relevant if a further discussion on what features explains better a data harmonization or on visual cues on how data harmonization improves classification tasks.


**Weaknesses:**

Weaknesses
The paper does not present a novel algorithm, and the comparison of existing method may be considered as expected results. Comparing a linear model with a strong hypothesis on prior distributions with a non-linear cyclegan may be considered unfair. The conclusion recommends, as perhaps expected, to use the non-linear method for data harmonization.


**Deanonymize Review:**

no

**Detailed Comments:**

Detailed Comments
* The study may be relevant if it provides a further analysis on what is essential in a data harmonization task. For instance, more details on what visual features or how distribution shifts could be interpreted as essential in a data harmonization. The paper as is, may be missing such elements. For instance, it does not contain any visual images of harmonized data.
* There are improvements over the vanilla ComBAT strategy, such as ComBAT-GAM, Pomonio et al, NeuroImage 2020. This may be relevant in a final comparison.
* CycleGANs are known to be in practice hard to train due to its unstable nature. A discussion on its training difficulty could help to better assess the complexity of using combat and a gans/cyclegans.
* The noise models should be further detailed. Exp-1,2 indicate that a gaussian intensity shift is centered in the mid-slice. Is this centering always fixed, or is this varying across data. If fixed, this may introduce a constant that can be easier to learn during data harmonization.
* Radiomic features should be further described. What aspects of medical images are reflected in them. Visual figures should be present in the main document.
* "table table", "figure figure"
* Fig 1 may benefit from having a quantitative measure, perhaps a classification score, to indicate that mid+right graphs shows improvements in site-independence.
* Is the Welch or Pearson's test being used?


**Justification Of The Preliminary Rating:**

* Recommendation for Weak Reject
* Relevance of comparative study on combat-cyclegan in the field should be strengthened.
* Comparison of linear vs non-linear model may be unfair (perhaps compare with the non-linear Combat-GAN)
* Results may be considered as expected results.

**Paper Type:**

validation/application paper

**Special Issue:**

no

---

### Official Review · AnonReviewer3 · 2021-03-09

**Confidence:** 4
**Preliminary Rating:** 2
**Final Rating:** 2

**Summary:**

This paper proposes a comparison of two promising methods for the harmonization of multi-center MR images: comBat and CycleGAN. To evaluate and compare both algorithms, the authors have designed five experiments evaluating the classification accuracy of an SVM model on radiomic features computed from the harmonized images of both evaluated methods. These experiments test the ability of comBat and CycleGAN to remove global noise, preserve manually added synthetic lesions, remove site effects and improve ASD patients and controls classification. Furthermore, statistical tests are performed on radiomic features to test for site affiliation and age correlation. The authors conclude that both methods can remove unwanted site effects while preserving biological information, but deep learning methods can outperform statistical methods like comBat.

**Strengths:**

The paper includes multiple experiments that test for a broad range of important scenarios while considering harmonization algorithm.

The authors designed five relevant experiments evaluating noise removal but also lesions preservation, site effects removal and disease classification.

Moreover, the authors use a public dataset to perform their evaluation which is important for the reproducibility of their work and further comparisons.

Finally, relevant statistical tests were conducted to better support the significance of their results.

**Weaknesses:**

The paper has multiple weaknesses:

First, the training procedure is not well defined. For instance, the authors did not mention how the 3D MRI data were preprocessed to fit in their 2D CycleGAN model nor how they split the data for cross-validation (i.e. are the folds computed slice wise or subject wise). Furthermore, the CycleGAN model that was used for comparison with comBat has been substantially modified from the original CycleGAN implementation. The impact of all these changes have not been discussed in the paper and I wonder: What is the value of comparing a custom architecture that has not been extensively tested?

In addition, the paper lacks direct evaluation of the harmonized images. The authors never visually show the harmonized images nor provide metric like PSNR to assess the noise reduction between the two methods. Also, it is not clear how the CycleGAN model that is proposed is supposed to remove the noise. If the input images of both sites have synthetic noise, what part of the objective is expected to reduce that noise?

Furthermore, the training setup of the experiments are not well defined. The amount of noise added to the images is not mentioned. What are the parameters of the Gaussian filters? The lack of details in the training procedures/parameters makes it hard to either reproduce the presented results or evaluate them.

**Deanonymize Review:**

no

**Detailed Comments:**

* The authors argue that CycleGAN does not use any prior hypothesis to model voxel intensities but in the case of their presented model, the added loss term “Dloss” assumes that the intensity of an image is similar in domain A and B. Otherwise, what is the use of this loss term ? That claim should be revised according to their CycleGAN proposition.

* I strongly recommend adding images to compare the visual performance of both algorithms. It would be interesting to see how well the noise is removed and how the synthetic lesions look like before and after harmonization for the two methods.

* The very last sentence of the conclusion is way too direct. Are you sure that CycleGAN is the best method to perform multi-center study? This claim should be reduced.

* Minor: In the results section there are some typos “Figure Figure 1… Figure Figure 2… Figure Figure 3…”

* Minor: The title says comBat but the method is further referred as ComBat in the text.

**Final Rating Justification:**

I appreciate the authors' efforts to address my comments and improve their paper. However, remaining concerns have made me keep my initial evaluation.

**Justification Of The Preliminary Rating:**

While the authors propose various experiments to compare comBat and CycleGAN, it lacks multiple training details and only demonstrate indirect evaluation of the two methods by relying on SVM classifiers in all their experiments. It is not clear why the authors proposed a custom CycleGAN model in a validation paper nor how the modification to the original architecture could affect their conclusions. Moreover, there is no images to support the authors claims. How do the images look like before and after harmonization?

**Paper Type:**

validation/application paper

**Questions To Address In The Rebuttal:**

* How were the 2D slices extracted from the images? Were they taken along a specific dimension? Did you consider every slice for every subject or selected specific slices?

* How was the cross-validation split performed? Did you first split on the subject and then compute the slices or compute the patches and split on the slices?

* What is the value of comparing a custom implementation of CycleGAN vs using a standard CycleGAN architecture following the implementation in (Zhu et al. 2017)? Could the conclusion be different if using the standard implementation?

* Difference in acquisition sites can drastically affect the intensity range of MRI images. Have the data been rescaled to be in the same intensity range, e.g. [0, 1] range, before training the models?

* Are the intensity distributions of the two sites before harmonization similar or drastically different? How do they look like after harmonization?

* In Exp 3. How is the mean and variance supposed to validate the geometrical integrity of a lesion? One could create slices with the same intensity mean and variance with drastically different geometry.

**Special Issue:**

no

---

### Meta-Review · Area_Chair1 · 2021-03-26

**Recommendation:** Accept (Poster)

**Metareview:**

This paper presents comprehensive experiments with two popular approaches to MRI harmonization: comBat and CycleGAN. Reviewers appreciated the extent of the experiments, but also had concerns about lack of: certain details; evaluation metrics;  comparison; and novelty. The first three have been addressed to some extent in the rebuttal (the authors edited the pdf but did not write rebuttals, which was a bit weird). The fourth may not be *that* important in a validation paper.

Even after the rebuttal stage, there are more negative than positive feelings among the reviewers towards the paper. However, it is the second best paper in my stack, so I will recommend acceptance as poster, with the understanding that this decision may be overruled by the program chairs.


**Paper Type:**

validation/application paper

---

### Decision · Program_Chairs · 2021-03-31

**Decision:**

Reject

**Comment:**

Despite encouraging experiments there are two many concerns and negative comments from reviewers and the area chair. In light of the overall score distribution the program committee decided to reject the paper and encourage the authors to submit a short paper.